# Genetic Associations of Novel Behaviour Traits Derived from Social Network Analysis with Growth, Feed Efficiency, and Carcass Characteristics in Pigs

**DOI:** 10.3390/genes13091616

**Published:** 2022-09-08

**Authors:** Saif Agha, Simon P. Turner, Craig R. G. Lewis, Suzanne Desire, Rainer Roehe, Andrea Doeschl-Wilson

**Affiliations:** 1The Roslin Institute, University of Edinburgh, Easter Bush, Edinburgh EH25 9RG, UK; 2Animal Production Department, Faculty of Agriculture, Ain Shams University, Cairo 11241, Egypt; 3Animal and Veterinary Sciences Department, Scotland’s Rural College, West Mains Road, Edinburgh EH9 3JG, UK; 4PIC, C/Pau Vila no. 22, Sant Cugat del Valles, 08174 Barcelona, Spain

**Keywords:** social network analysis, genetic parameters, welfare, aggressiveness, pigs

## Abstract

Reducing harmful aggressive behaviour remains a major challenge in pig production. Social network analysis (SNA) showed the potential in providing novel behavioural traits that describe the direct and indirect role of individual pigs in pen-level aggression. Our objectives were to (1) estimate the genetic parameters of these SNA traits, and (2) quantify the genetic associations between the SNA traits and commonly used performance measures: growth, feed intake, feed efficiency, and carcass traits. The animals were video recorded for 24 h post-mixing. The observed fighting behaviour of each animal was used as input for the SNA. A Bayesian approach was performed to estimate the genetic parameters of SNA traits and their association with the performance traits. The heritability estimates for all SNA traits ranged from 0.01 to 0.35. The genetic correlations between SNA and performance traits were non-significant, except for weighted degree with hot carcass weight, and for both betweenness and closeness centrality with test daily gain, final body weight, and hot carcass weight. Our results suggest that SNA traits are amenable for selective breeding. Integrating these traits with other behaviour and performance traits may potentially help in building up future strategies for simultaneously improving welfare and performance in commercial pig farms.

## 1. Introduction

Mixing unfamiliar pigs is a common procedure in commercial farms, although it usually results in unstable social structures and increased aggressive behaviour in the pen. The negative consequences of this procedure on the welfare, performance, and health of pigs have been reported in several studies [1,2,3]. Existing management methods have limitations in controlling mixing aggression on commercial farms and are poorly adopted [4]. Breeding could complement the ongoing efforts to find effective management changes to provide a permanent solution at a relatively low cost to reduce aggressive behaviour in pigs [5]. However, it is critical to identify the desirable behavioural traits that could effectively quantify the role that individuals play within the complex social structure of aggressive behaviour in the pen in order to choose the optimum traits for selection [6,7]. Skin lesions are considered valuable proxy traits for breeding due to their moderate heritability and positive genetic correlation with several aggressive behaviour traits [8,9]. For example, selection against anterior skin lesions at 24 h post-mixing is expected to effectively decrease aggressive behaviour at mixing [10]. However, skin lesions do not capture a significant part of the variance in aggressive interactions and do not take into account the opponents causing the lesions, or previous interactions between the same opponents [5,11]. Therefore, to quantify the aggressive behaviour of an individual for selection purposes, it is important to consider its relative role in the aggressive network structure of the group.

Social network analysis (SNA) has increased in popularity in the fields of animal ethology and welfare science as it provides new perspectives regarding complex social structures [12,13]. In pigs, SNA has been used to monitor and understand the patterns of different harmful behaviours such as fighting, bullying, and tail biting [14,15,16], as well as identifying the dominance hierarchy in the group [16]. Moreover, SNA has been shown to provide a more accurate prediction of skin lesions resulting from long-term chronic aggression compared to conventional approaches, which make no reference to the opponents fought [17]. Besides describing the group-level social dynamics, SNA has the potential to quantify novel individual behavioural traits that describe the direct and indirect role of each animal in the social structure, which could not be identified through traditional approaches such as summing the number of dyadic fights the animal engaged in [18,19,20]. These individual SNA traits have shown the capacity to identify the key individuals that have a major influence on the aggressive interactions and injuries in the pen [21]. Moreover, the individual SNA traits were found to be heritable in pigs (h^2^ = 0.09–0.26) [22]. This would suggest that these traits are amenable for selective breeding. Furthermore, selecting for an index that combines several SNA traits was expected to decrease skin lesions in pigs at different parts of the body, after mixing into new social groups and thereafter in stable group conditions, compared to selecting for each SNA trait separately [22]. However, aggressive behaviour performed when regrouped with unfamiliar animals serves the purpose of establishing dominance relationships, which may determine prioritised access to limited resources such as food [23]. Therefore, it is crucial to examine the relationship between aggressive behaviour and performance traits in order to confirm that it is possible to improve the performance and welfare of the pigs simultaneously. The association between SNA and performance traits in pigs was previously investigated, but only at the phenotypic level [18], and the genetic association between these traits is still unknown. Therefore, in this study, we applied the social network approach on aggressive behaviour data in order to (1) estimate the genetic parameters of the SNA traits in pigs and (2) quantify the phenotypic and genetic associations between the SNA traits and performance traits.

## 2. Materials and Methods

### 2.1. Animals

This study uses the same dataset as described in Desire et al. (2015) [24]. Briefly, the study was conducted on 900 pigs (450 females and 450 castrates) from a PIC commercial herd reared in the USA between December 2012 and June 2013. The animals were the progeny of 116 sires and 391 dams, and the pedigree information used also included the grandparents (the total number of animals in the pedigree was 4104). Pigs from eight different terminal genetic lines were available, which were based on crosses of two maternal lines and seven sire lines. Single-sex groups of 18 pigs of mixed genetic lines were formed by mixing nine pigs from two nonadjacent weaning pens, and the total number of groups was 50. Each pen consisted of pigs from an average of 11.6 (SD = 2.1) litters, and the average number of pigs per litter per pen was 1.5 (SD = 0.81). The average age at mixing was 69 days (SD = 5.2). Pens that were mixed on the same day were considered the same batch (nine batches in total). The animals were fed ad libitum on a dry pelleted food. The floor space allowance per animal was 0.65 m^2^. 

### 2.2. Behavioural Traits

The animals were observed via video recording for 24 h post-mixing. The duration of the reciprocal fighting behaviour between each pair of animals and their identities were registered. Reciprocal fighting behaviour was defined as a fight that lasted more than one second where both pigs were involved in pushing, head knocking, or biting. Pushing was defined as using the head or shoulder to move the other pig aside by applying pressure, head knocking was defined as the rapid swinging of the head to deliver a blow, and biting was defined as opening the mouth and delivering a bite that contacted and injured the other pig. Three observers used time-lapse video equipment to extract the duration of each behavioural bout to the nearest second. The analysis of three 1-h samples of data showed a significant degree of inter-observer agreement (r = 0.83, *p* < 0.001) [25].

### 2.3. SNA Traits 

SNA was performed using the “*igraph*” package in the R software [26]. SNA transforms the numerical data of the aggressive interactions between animals into graphs, where the animals are displayed in term of nodes. The nodes are connected through edges, i.e., lines, which represent the interaction between animals. In this study, the edges were weighted based on the total duration of the reciprocal fighting behaviour between each pair of animals. SNA was used to give a score, i.e., traits, for each animal to describe its role in the network, i.e., pen [20,27]. Here, we considered the SNA traits that were found to have a significant effect on skin lesions based on the previous studies [17,18,22]. These included degree centrality, weighted degree centrality, betweenness centrality, closeness centrality, eigenvector centrality, and clustering coefficient, which were calculated using the weighted social networks. The definition of each trait is shown in Table 1. An example of a network is shown in Figure 1. All of these SNA traits showed considerably skewed distributions; therefore, a square root transformation was applied to approach a normal distribution.

### 2.4. Performance Traits

For each animal, the average daily gain on test, i.e., test daily gain (TDG) was calculated for the performance test period from 70 to 172 days of age (877 g/d, SD 120.17), and during the entire life, the lifetime daily gain (LDG), which was calculated by dividing the off-test body weight by the days of age (685 g/d, SD 76.76). Feed intake data were registered for each animal using single-space electronic feed intake recording equipment (FIRE; Osborne Industries, Osborne, KS, USA) every alternate 2 weeks throughout the finishing period (i.e., the phase in which pigs are fed until they reach market weight), which was approximately from 70 to 172 days of age in the current experiment. In the intervening weeks, when the animals did not have access to FIRE feeders, feed was provided via a multi-spaced trough. Feed efficiency (FE) was calculated by dividing the TDG by daily feed intake (DFI). The pigs were weighed, i.e., final body weight (FBW), and slaughtered at an average age of 178 d (SD 4.6) in a commercial abattoir. Hot carcass weights (HCW) in kg were recorded, and back fat (BF) in mm and loin depth (LD) in mm were measured on the carcass at the tenth rib using Fat-O-Meter equipment (SFK, Hvidovre, Denmark).

The phenotypic associations between SNA and performance traits were calculated using Spearman rank correlations using the R software [28].

### 2.5. Genetic Parameter Estimates

The genetic variance components and the genetic associations of all transformed SNA traits and the performance traits were estimated through a series of univariate and bivariate analyses using the following linear animal model:y = Xb + Za + Wc + e
where y is the vector of records for the SNA traits or performance traits, and X, Z, and W are the incidence matrices of the fixed effects, random genetic effects, and environmental (pen) effects, respectively. Vectors b, a, c, and e represent the fixed effects, additive direct genetic effects, common environmental pen effects, and residual error, respectively. The fixed effects included the genetic lines (nine levels), sex (two levels: female, and castrated males) and batch (nine levels; groups mixed on the same day were classed as the same batch), and the body weight at time of mixing was fitted as a covariate. 

The models’ parameters were estimated by Bayesian statistics using the Markov Chain Monte Carlo (MCMC) method of Gibbs sampling utilizing the BLUPF90 family software [29]. Flat priors were assumed for the systematic effects and the covariance components. The conditional prior distributions of the random effects a, c, and e were sampled from multivariate normal distributions (N), as follows: P(a|A, G^0^) ~ N(0, A ⊗ G^0^)
P(c|C^0^) ~ N(0, *I* ⊗ C^0^)
P(e|R^0^) ~ N(0, *I* ⊗ R^0^)
where A is the additive genetic relationship matrix, G^0^ is the additive genetic (co)variance matrix, *I* is the identity matrix, and ⊗ represents the Kronecker product of the matrices. C^0^ is the (co)variance matrix of the environmental pen effects and R^0^ represents the (co)variance matrix of the residuals. The conditional posterior distributions of the covariance components of G^0^, C^0^, and R^0^ were sampled from inverse Wishart distributions. The posterior distribution of the genetic and environmental effects was derived from a Markov chain of 1,000,000 iterations of the parameters, where the burn in of 100,000 was applied to obtain a stationary MCMC distribution, and a lag of 20 iterations was applied to reduce the autocorrelations between consecutive iterations. The convergence of the Markov chain was tested using the algorithms of [30]. The posterior means and the standard deviations of the marginal distributions of the genetic and environmental parameters of SNA and performance traits were estimated, in addition to the 95% highest posterior density intervals (HPD95%).

## 3. Results

The descriptive statistics of the transformed SNA traits and performance traits are shown in Table 2 and Appendix A. Considerable phenotypic variation was observed for the SNA traits.

### 3.1. Heritability

The heritability estimates for all SNA traits were significant, the 95% highest posterior density intervals (HPD95%) did not include zero, and ranged from 0.01 for closeness centrality to 0.35 for the weighted degree centrality (Table 3). The posterior means of the common environmental pen effect for closeness centrality was the highest amongst the different SNA traits (c^2^ = 0.80).

### 3.2. Genetic and Phenotypic Correlations

The posterior means for the genetic correlations and the HPD95% for SNA traits are presented in Table 4. Positive genetic correlations, with the HPD95% above zero, were observed between degree centrality and weighted degree centrality, and eigenvector centrality. The corresponding phenotypic correlations between these traits were also moderate to strong (r > 0.41) (Appendix A). In contrast, closeness centrality showed negative genetic correlations, with the HPD95% above zero, with weighted degree centrality and eigenvector centrality; however, these traits were positively correlated at the phenotypic level. Furthermore, the clustering coefficient showed high negative genetic correlations with the rest of the SNA traits (rg < −0.88), although positive correlations were observed between these traits at the phenotypic level.

The posterior means of the genetic correlations and the 95% highest posterior density intervals (HPD95%) between SNA and performance traits are shown in Table 5. The estimates of the genetic correlations between degree, weighted degree, eigenvector centrality, clustering coefficient, and performance traits were non-significant, with HPD95% including zero, except for the weighted degree and BF, which showed a high and negative genetic correlation. On the other hand, negative genetic correlations were observed between betweenness centrality, closeness centrality, and TDG, FBW, and HCW, with the HPD95% above zero. Furthermore, negative genetic correlations, with the HPD95% above zero, were observed between betweenness centrality and LDG. At the phenotypic level, low correlations were observed between these traits (Appendix A). Furthermore, moderate to large correlations were estimated between the clustering coefficient and all analysed performances traits as well as between FE and eigenvector centrality, although this showed a lack of creditability based on the HPD95% and therefore requires further studies to identify the relevance. 

## 4. Discussion

### 4.1. Heritability

In previous studies, SNA showed the potential, compared to conventionally recorded dyadic interactions and skin lesions traits, to provide novel behaviour traits that describe the direct and indirect role of each animal in social interactions [16,17]. However, only one study has estimated the genetic parameters of these SNA traits [22] and none have explored the genetic association with performance traits. In the current study, the heritability estimates of the SNA traits were low to moderate. Although these estimates were derived from social networks that were weighted by the duration of reciprocal fights, these estimates are similar to the heritabilities of SNA traits previously estimated in pigs from unweighted social networks (h^2^ ranged from 0.09 and 0.26) [22]. Furthermore, these heritability estimates found in pigs are in the same range as the heritabilities of SNA traits for other forms of social interaction in other species, e.g., rhesus macaques [31], marmots [32], and humans [33]. This would suggest that the SNA traits are valuable candidate behaviour traits that could be used for selective breeding. 

Closeness centrality showed the highest common environmental effect amongst the SNA traits in this study (c^2^ = 0.80). A similar result was also found previously (c^2^ = 0.59) [22]. Closeness centrality is a trait that measures how close an animal is to all other animals in the group, therefore, it is highly dependent on the social structure in the pen. This could be the reason for the low direct genetic effect found for this trait [20,22].

### 4.2. Genetic and Phenotypic Correlations

High positive genetic correlations were observed between the eigenvector centrality and degree and weighted degree centralities (rg > 0.87). Similar high positive genetic correlations were also observed between these traits previously in pigs (rg > 0.88) [22]. However, in the current study, eigenvector centrality showed a negative genetic correlation with closeness centrality, although a high positive genetic correlation was reported in Agha et al. (2022) [22]. One difference between these studies is that the SNA traits in the current study were calculated from social networks that were weighted by the duration of reciprocal fights between animals, while the SNA traits were calculated from unweighted social networks in Agha et al. (2022) [22]. That may indicate that weighting the edges between nodes, using the duration of reciprocal fights, may change the score of some SNA traits and consequently the associations between these traits. However, it should be considered that closeness centrality had a low heritability and high common environmental effect estimates, therefore, the genetic associations between these traits and other SNA traits may require further investigation.

On the other hand, a negative genetic correlation was observed between weighted degree and closeness centrality, although a positive phenotypic correlation was observed between these traits. Moreover, the clustering coefficient showed strong negative genetic correlations with other SNA traits, which is in agreement with the results found previously in pigs in unweighted social networks [22]. However, in this study, the clustering coefficient showed positive correlations with other SNA traits at the phenotypic level. The genetic correlations can be opposite to the observed phenotypic correlations if the environmental influence is sufficiently strong in the other direction [34,35]. By definition, both clustering coefficient and closeness centrality are two traits that depend largely on the dynamics of the social group (Table 1). Therefore, the score for these traits is highly dependent on the social structure of the group, and weighting interactions by duration of reciprocal fighting may also have affected the score for these traits. These could be reasons for the difference between the phenotypic and genetic correlations found for these two traits in the current study. 

The aggressive behaviour of pigs is performed at regrouping functions to establish dominance relationships that can subsequently affect access to limited resources [23]. Hence, aggressive behaviour could have a direct effect on performance traits via feed intake. Understanding the effect of aggressive behaviour on performance traits is essential for simultaneously improving the performance and welfare of pigs. At the phenotypic level, Agha et al. (2020) [18] investigated the associations between SNA and performance traits in Duroc pigs. They found low correlations between centrality traits and feed conversion ratio and average daily consumption, while non-significant associations were found between SNA and growth traits. However, to the best of our knowledge, the current study is the first attempt to estimate the genetic associations between SNA and performance traits in farm animals. Here, we found non-significant genetic correlations, i.e., HPD95% including zero, between degree centrality, eigenvector centrality, clustering coefficient, and performance traits. Furthermore, low correlations were observed between these traits at the phenotypic level. These results are in line with the results observed in Agha et al. (2020) [18]. Pigs can adapt their feeding behaviour to maintain feed intake without engaging in unnecessary fights when feed is available throughout the day [36]. That could explain the low phenotypic correlations between the SNA and performance traits observed in these populations, where feeding was ad libitum. However, this may not be the case when feeding is restricted or housing conditions are different [37,38]. 

In this study, high negative genetic correlations were found between betweenness centrality and closeness centrality on the one hand and TDG, FBW, and HCW on the other. Additionally, betweenness centrality showed a high negative genetic correlation with LDG. At the phenotypic level, low correlations were observed between these traits. In the same pig population, positive genetic correlations were observed between daily gain and skin lesions 24 h post-mixing [24]. Furthermore, high positive genetic correlations were observed between betweenness centrality, closeness centrality, and skin lesions 24 h post-mixing in a different pig population [22]. Therefore, combining the results of these studies would suggest that the animals that have a central location in the pen network, as indicated by their linking disconnected subgroups or being close to other animals, receive more skin lesions compared to other animals in the pen, and that this either directly (through social stress) or indirectly (through compromised access to feed) could affect their daily gain, final body weight, and carcass weight. 

### 4.3. Implications for Breeding

The genetic improvement of animal welfare in commercial farms is challenging. Monitoring the behaviour of the animals within a group has, until now, been laboriously conducted by human experts [39]. However, due to the recent advances and reduction in the cost of monitoring technology, there is an increasing interest in installing cameras and sensors on farms to monitor the behaviour of the animals throughout the day [40]. In the future, these cameras are expected to provide high-resolution records that can be automatically analysed to detect the position and the interaction of the animals in the pen [41]. Such automated systems will provide valuable data that can be integrated with SNA methods to automatically extract behavioural traits that identify the role of each animal in the social network. 

The genetic parameters of the SNA traits estimated in this study, and the study of Agha et al. (2022) [22], would suggest that these traits are amenable for selective breeding. The selection of an index that combines social network traits, i.e., the eigenvector-clustering index, showed the potential in decreasing the injuries and benefitting the group in the short-term and long-term compared to the selection of eigenvector centrality and clustering coefficient separately (Agha et al. 2022) [22]. Therefore, combining these results with the observed low correlations between these SNA traits and performance traits in the current study would suggest that the selection of an index that considers SNA traits would decrease the injuries in the group without having a considerable effect on the performance of the pigs. However, these studies are only the first attempts towards the integration of SNA traits in breeding strategies for improving the performance and welfare of pigs. Further investigations are still needed to clarify important questions such as the effect of group sizes on the score of the SNA traits. It would also be interesting to investigate whether the same pig will have the same centrality score, i.e., position within the network, during different life stages. This information would help to effectively integrate the SNA traits with other behaviour and performance traits to establish a selection index that simultaneously improves the performance and welfare of pigs under different commercial farm conditions.

## 5. Conclusions

Controlling the aggressive behaviour in commercial pig farms requires suitable traits that describe the role of each animal in the aggression. In this study, social networks were performed and weighted using the duration of reciprocal fights between animals, and novel behaviour traits were extracted, i.e., SNA traits. Our results suggest that these SNA traits are heritable and genetically correlated. The genetic correlations of the SNA with performance traits were not significantly different from zero, except for weighted degree and HCW, and between both betweenness and closeness centrality and DFI, FBW, and HCW. Considering SNA traits alongside performance traits may help to build up future strategies for simultaneously improving the performance and welfare in modern commercial pig farms. Finally, the current study can be considered a case study for a novel approach that could be applied to other animal species and different forms of social interaction to better understand the social structure of different animal groups.

## Figures and Tables

**Figure 1 genes-13-01616-f001:**
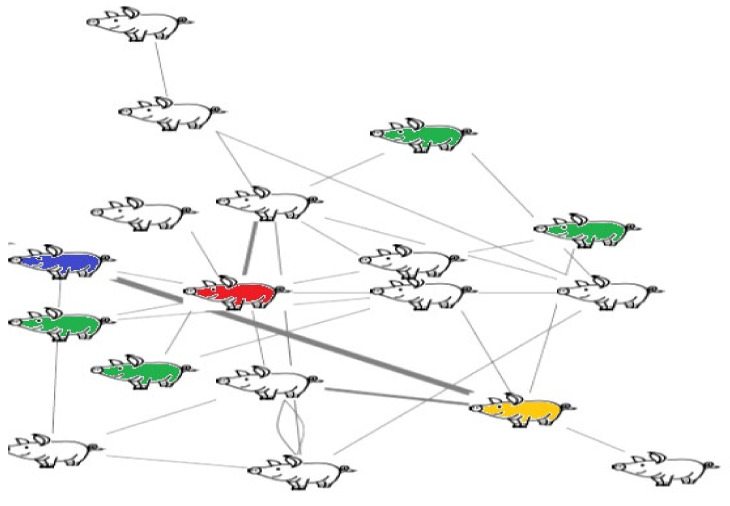
Example of a network. The pigs display the nodes, and the arrows display the edges. The thickness of the arrows represents the weight, i.e., the duration of the reciprocal fight. Red pig has the highest degree centrality, betweenness centrality, and closeness centrality in the pen. Yellow pig has the highest weighted degree centrality in the pen. Blue pig has the highest eigenvector centrality in the pen. Green pigs have the highest clustering coefficient in the pen.

**Table 1 genes-13-01616-t001:** Definition of the social network analysis traits considered in this study.

Measures	Definition	Interpretation
Degree centrality	The number of edges attached to a node.	The number of animals that a particular animal directly engaged with.
Weighted degree centrality	The sum of weights associated with every edge incident to the corresponding node.	The sum of the duration of the reciprocal fights that the focal animal was involved in.
Betweenness centrality	The number of shortest paths that pass through the considered node.	Measures the importance of the animal in connecting different subgroups of the pen engaging in aggression.
Closeness centrality	The average of the shortest path length between that node and all other nodes in the network.	Measures how ‘close’ an animal is to all other animals in a pen in terms of engaging in aggression. Animals that engage in aggression directly with many of their pen mates have high closeness centrality.
Eigenvector centrality	The connectivity of a node within its network, according to the degree centrality of the node and the degree centrality of the nodes that it connects with.	Takes into consideration both the degree centrality of the focal individual and the degree centrality of its opponents.
Clustering coefficient	The proportion of an individual node’s connections that are also directly connected with each other relative to the number of theoretically possible connections.	Quantifies what proportion of animals that the focal individual directly engages with also interact with each other, relative to the number of all possible aggressive interactions.

**Table 2 genes-13-01616-t002:** Descriptive statistics of the transformed SNA traits and performance traits.

Category	Trait	Mean	SD	Max	Min
SNA	Degree centrality	0.12	0.13	1.00	0
	Weighted degree centrality	2.30	5.92	58.4	0
	Closeness centrality	0.02	0.02	0.08	0
	Eigenvector centrality	0.17	0.31	1.00	0
	Betweenness centrality	0.06	0.10	0.63	0
	Clustering coefficient	0.09	0.21	1.00	0
Performance	TDG	887.5	117.4	1198.1	523.4
	LDG	695.9	75.10	889.5	435.2
	DFI	2.28	0.29	3.12	1.38
	FE	0.003	0.001	0.004	0.002
	FBW	120.11	12.03	155.00	84.00
	HCW	94.10	8.97	127.57	67.19
	BF	17.97	4.34	33.10	7.10
	LD	62.24	8.88	89.30	35.90

TDG = Test daily gain (g/d), LDG = lifetime daily gain (g/d), DFI = daily feed intake (g/d), FE = feed efficiency, FBW = final body weight (kg), HCW = hot carcass weight (kg), BF = back fat (mm), LD = loin depth (mm).

**Table 3 genes-13-01616-t003:** Posterior means of heritability (h^2^), the phenotypic proportions of the variance due to the environmental pen effects (c^2^), the phenotypic variances (Vp), and the 95% highest posterior density intervals (HPD95%) for social network analysis traits for aggressive behaviour.

Trait	h^2^	HPD95%		c^2^	HPD95%		Vp	HPD95%	
Degree	0.13	0.016	0.276	0.27	0.118	0.439	0.020	0.016	0.026
Weighted degree	0.35	0.013	0.651	0.10	0.000	0.248	50.91	37.06	67.64
Betweenness centrality	0.17	0.003	0.417	0.27	0.124	0.426	0.026	0.020	0.031
Closeness centrality	0.01	0.000	0.029	0.80	0.724	0.871	0.003	0.002	0.004
Eigenvector centrality	0.10	0.003	0.283	0.12	0.015	0.231	0.121	0.102	0.141
Clustering coefficient	0.04	0.003	0.172	0.14	0.000	0.290	0.047	0.035	0.060

**Table 4 genes-13-01616-t004:** Posterior means of the genetic correlations and the 95% highest posterior density intervals (HPD95%) for social network analysis traits for aggressive behaviour ^1^.

Trait	Weighted Degree	Betweenness Centrality	Closeness Centrality	Eigenvector Centrality	Clustering Coefficient
Degree	0.33 (−1, 0.92)	0.59 (−0.80, 1)	0.55 (−0.69, 1)	**0.92** (0.57, 1)	−0.73 (−1, 0.65)
Weighted degree		0.10 (−0.97, 0.71)	**−0.80** (−1, −0.25)	**0.87** (0.42, 1)	**−0.92** (−1, −0.59)
Betweenness			0.78 (−0.03, 1)	0.79 (−0.27, 1)	−0.55 (−1, 0.56)
Closeness				**−0.97** (−1, −0.84)	−0.70 (−1, 0.80)
Eigenvector					**−0.95** (−1, −0.73)

^1^ Bold font signifies correlation estimates with HPD95% that did not include zero.

**Table 5 genes-13-01616-t005:** Posterior means of the genetic correlations and the 95% highest posterior density intervals (HPD95%) between social network analysis traits and performance traits ^1^.

Trait	Degree	Weighted Degree	Betweenness Centrality	Closeness Centrality	Eigenvector Centrality	Clustering Coefficient
TDG	−0.30 (−1, 0.96)	−0.05 (−0.95, 0.78)	**−0.87** (−1, −0.57)	**−0.70** (−1, −0.07)	0.31 (−0.40, 0.99)	−0.79 (−1, 0.37)
LDG	−0.32 (−1, 0.95)	−0.05 (−0.89, 0.76)	**−0.86** (−1, −0.50)	−0.61 (−1, 0.18)	0.18 (−0.61, 0.98)	−0.74 (−1, 0.59)
DFI	−0.14 (−1, 0.94)	−0.15 (−0.95, 0.53)	−0.37 (−0.96, 0.46)	−0.63 (−1, 0.22)	−0.43 (−1, 0.26)	−0.83 (−1, 0.05)
FE	−0.07 (−1, 0.80)	−0.08 (−1, 0.71)	0.34 (−0.47, 0.99)	0.05 (−0.89, 0.99)	−0.61 (−1, 0.06)	−0.72 (−1, 0.35)
FBW	−0.30 (−1, 0.96)	−0.12 (−0.94, 0.57)	**−0.87** (−1, −0.57)	**−0.74** (−1, −0.17)	0.31 (−0.40, 0.99)	−0.78 (−1, 0.38)
HCW	−0.31 (−1, 0.93)	−0.20 (−0.95, 0.50)	**−0.69** (**−1**, **−0.05**)	**−0.77** (**−1**, **−0.24**)	0.18 (−0.62, 0.94)	−0.81 (−1, 0.16)
BF	−0.16 (−1, 0.90)	**−0.63** (**−0.97**, **−0.08**)	−0.59 (−0.97, 0.12)	0.71 (−0.08, 1)	−0.26 (−0.93, 0.36)	−0.76 (−1, 0.28)
LD	−0.13 (−1, 0.94)	0.23 (−0.70, 0.92)	0.03 (−0.65, 0.94)	0.25 (−0.85, 1)	0.44 (−0.22, 0.99)	−0.79 (−1, 0.13)

TDG = Test daily gain (g/d), LDG = lifetime daily gain (g/d), DFI = daily feed intake (g/d), FE = feed efficiency, FBW = final body weight (kg), HCW = hot carcass weights (kg), BF = back fat (mm), LD = loin depth (mm). ^1^ Bold font signifies correlation estimates with HPD95% that did not include zero.

## Data Availability

Data can be made available upon request.

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
