# Peer review of "Genetic Associations of Novel Behaviour Traits Derived from Social Network Analysis with Growth, Feed Efficiency, and Carcass Characteristics in Pigs"

_genes, 2022, doi:10.3390/genes13091616_

Round 1

Reviewer 1 Report

Thanks for making efforts on conducting the analysis. The manuscript is well-written and easy to ready through. Few minor questions and comments are listed below:

Materials and Methods:

1. What are the age range for pigs when grouping (min and max)? Are those pigs in the same pen have similar age? How about weight variation? 

2. Line 145 (Model): The common litter effect is not modeled. Has the author tried to fit litter effect? 

3. Can the authors provide the descriptive statistics and distribution plots in the supplement for SNA in non-transformed scale? 

4. Line 343. "Optimizing the aggressive behaviour..." Is "Optimizing" the right wording? 

5. Line 351. "Considering SNA traits alongside performance traits may help in building up fu- 350 ture strategies for optimizing behaviour to improve modern commercial pig farms." Please clarify to improve what aspect.

6. The authors use the traditional animal model with direct genetic effect. Can those SNA traits be analyzed with the social indirect genetic model (including both direct and indirect genetic effects)? Please discuss this.

7. Most of the genetic correlation among traits are non-significant with HPD95% included zero. Please discuss more on possible reasons and suggestion on improvement.

8. I agree with the authors on the comment that different social structure/group size might affect the SNA results. I look forward to see any validation and prediction study on this. 

Author Response

Response to the reviewer

We would like to thank the reviewer for the valuable comments. Please see below our response to each question (In blue).

Thanks for making efforts on conducting the analysis. The manuscript is well-written and easy to ready through. Few minor questions and comments are listed below:

Materials and Methods:

  1. What are the age range for pigs when grouping (min and max)? Are those pigs in the same pen have similar age? How about weight variation? 

The minimum age is 55 days, and the maximum is 110 days. The average age at mixing was 69 days (SD = 5.2), included in Line 89.

The minimum weight was 20kg and the maximum weight was 45kg. Please note that the body weight at mixing was included in the model as a covariate (Line 151).

  1. Line 145 (Model): The common litter effect is not modeled. Has the author tried to fit litter effect? 

We tried the model with and without including the common litter effect, and we obtained similar genetic parameters estimates. Furthermore, the used model, in this study, is similar to the model used by Desire et al, 2015, on the same data, to estimate the genetic parameters of skin lesions and their associations with production traits.

  1. Can the authors provide the descriptive statistics and distribution plots in the supplement for SNA in non-transformed scale? 

Based on the reviewer suggestion, we added a table for the descriptive statistics in the supplementary material as the referee suggested. Please see table S1 in the supplementary material and Line 185 in the main text. By including both the descriptive statistics’ tables (Table 2 and Table S1) for the transformed and non-transformed SNA, we believe that the reader will have the relevant information regarding the SNA traits of this study.

  1. Line 343. "Optimizing the aggressive behaviour..." Is "Optimizing" the right wording? 

We agree and the word has changed to “Controlling”.

  1. Line 351. "Considering SNA traits alongside performance traits may help in building up fu- 350 ture strategies for optimizing behaviour to improve modern commercial pig farms." Please clarify to improve what aspect.

The sentence has been modified to be “Considering SNA traits alongside performance traits may help in building up future strategies for simultaneously improving the performance and welfare in modern commercial pig farms”.

  1. The authors use the traditional animal model with direct genetic effect. Can those SNA traits be analyzed with the social indirect genetic model (including both direct and indirect genetic effects)? Please discuss this.

Thanks for the suggestion. This point in particular has been discussed in our previous paper published also in Genes (Agha et al, 2022). Therefore, we did not discuss further this point in the current study. Please see the paragraph from this paper below,

Estimates of correlations between direct and social genetic effects for dyadic aggressive behaviour were found to be positive, implying that selecting animals with low genetic propensity for engaging in aggressive behaviour may be beneficial for reducing aggression in the group as a whole. However, estimation of social genetic effects requires very large data and a particular data structure that may be difficult to obtain in commercial settings. In contrast, selection for SNA traits, which show similar heritability estimates as dyadic behavioural traits but intrinsically incorporate social interactions, may thus be a more efficient way to reduce aggressive behaviour at the pen level.

  1. Most of the genetic correlation among traits are non-significant with HPD95% included zero. Please discuss more on possible reasons and suggestion on improvement.

The reason for the non-significant correlation between SNA traits and performance traits was included in the Line 297. “Pigs can adapt their feeding behaviour to maintain feed intake without engaging in unnecessary fights when feed is available throughout the day [36]. That could explain the low phenotypic correlations between SNA and performance traits observed in these populations, where feeding was ad-libitum. However, this may not be the case when feeding is restricted or housing conditions are different [37,38].

  1. I agree with the authors on the comment that different social structure/group size might affect the SNA results. I look forward to see any validation and prediction study on this. 

We agree with reviewer, and we are planning to run new social network analyses, using different social structure and group sizes, to validate our results in the near future.

Reviewer 2 Report

This manuscript brings up interesting points in the intersection of social network analysis, genes, and animal management.  studies. The ideas and intention behind the project are important to understanding the responses and traits of animals to genetics and their management.

The overall manuscript is in good condition and well written, however I only have a few comments for added clarification.

Lns 87-89: I believe I'm not understanding terminology in this sentence, stating pens were formed by mixing 9 pigs with the same sex to create 50 groups. Does "9 pigs" refer to the terminal genetic lines, and if so how would breeding occur with same sex? Also if it refers to the parental level, then the "2 nonadjacent weaning pens" doesn't make sense. Apologies if I'm missing something, but I recommend rewording this part of the paragraph.

Lns 91-92: Again, I may not understand terminology here, but I'm not sure what 'batch' refers to.

Ln 97: define reciprocal fighting. I assume its a dyadic event, but the duration (<1sec) seems short for an initiator-recipient, recipient-initiator interaction to occur. Likewise, I would encourage the authors to include an ethogram of recorded behavior, to further clarify behavioral responses of the target animals. As currently written, the behavioral results are not reproducible.

lns 108-116: I haven't heard of SNA measures referred to as traits before, but believe the terminology is being used here as a reference to genetics. However, it may be more clear in the remainder of the manuscript to differentiate SNA 'measures with performance/physical 'traits'. Not a major issue, but I stumbled over each use so assume other readers would, as well.

 Lns 126-140: Lots of jargon in this paragraph, i.e., 'daily gain', 'on test', 'finishing period', 'feed efficiency, etc, which many people reading this journal may be unfamiliar with. Recommend defining or revising.

Ln 172: Does heritability refer to the genetic matrix described above, or was it calculated/defined in an undescribed manner? It's the first use of the term in the text, and I'm unsure how it fits into this section as it is currently used. Likewise, it appears to be used distinctly from genetic correlations in the remainder of the manuscript.

Lns 247-251: I'm glad authors noted the dependence of closeness centrality on social structure, particularly given the result - it was my first thought with the measure being included for analysis.

Lns 269-279: Adding to my point above, clustering coefficient and closeness centrality are dependent on social structure, which may not have developed sufficiently within the study window to find results. Happy the authors note this. I would also add here that weighted vs unweighted edges likely differ, but weighted based on duration, frequency, number of interactions, etc all change this measure, which again circles to my concern of referring to these SNA measures as 'traits'. Because we modify them using math, we need a lot more information before I would personally be comfortable aligning them with other, physical or biological traits.

Lns 307-312: Brilliant! Wonderful outcome and interpretation.

lns 320-323: Can we live in this world now!? I cant wait until we do.

Author Response

Response to the reviewer

We would like to thank the reviewer for the valuable comments. Please see below our response to each point (In blue). 

This manuscript brings up interesting points in the intersection of social network analysis, genes, and animal management.  studies. The ideas and intention behind the project are important to understanding the responses and traits of animals to genetics and their management.

The overall manuscript is in good condition and well written, however I only have a few comments for added clarification.

Lns 87-89: I believe I'm not understanding terminology in this sentence, stating pens were formed by mixing 9 pigs with the same sex to create 50 groups. Does "9 pigs" refer to the terminal genetic lines, and if so how would breeding occur with same sex? Also if it refers to the parental level, then the "2 nonadjacent weaning pens" doesn't make sense. Apologies if I'm missing something, but I recommend rewording this part of the paragraph.

Based on the reviewer suggestion, we modified the sentence as follow; “Pigs from 8 different terminal genetic lines were available, which were based on crosses of 2 maternal line and 7 sire lines. Single sex groups of 18 pigs of mixed genetic line were formed by mixing 9 pigs from 2 nonadjacent weaning pens, and the total number of groups were 50”.

Lns 91-92: Again, I may not understand terminology here, but I'm not sure what 'batch' refers to.

In Line 93, “Pens that were mixed on the same day were considered as the same batch (9 batches in total)”.

Ln 97: define reciprocal fighting. I assume its a dyadic event, but the duration (<1sec) seems short for an initiator-recipient, recipient-initiator interaction to occur. Likewise, I would encourage the authors to include an ethogram of recorded behavior, to further clarify behavioral responses of the target animals. As currently written, the behavioral results are not reproducible.

Based on the reviewer suggestion, we added the following sentence to this paragraph to clarify this point for the reader. Line 100. “Pushing was defined as using the head or shoulder to move the other pig aside by applying pressure, head knocking was defined as the rapid swinging of the head to deliver a blow and biting was defined as opening the mouth and delivering a bite which contacted and injured the other pig.”

Lns 108-116: I haven't heard of SNA measures referred to as traits before, but believe the terminology is being used here as a reference to genetics. However, it may be more clear in the remainder of the manuscript to differentiate SNA 'measures with performance/physical 'traits'. Not a major issue, but I stumbled over each use so assume other readers would, as well.

Thanks for the suggestion. Indeed, we defined the SNA measures as traits in line with our previous work (https://www.mdpi.com/2073-4425/13/4/561; https://www.mdpi.com/2076-2615/10/11/2123) and the definition of traits in the genetic context:  “a trait, as related to genetics, is a specific characteristic of an individual” (https://www.genome.gov/genetics-glossary/Trait). Although the trait values depend on the data and methods used to construct the social network, each of the resulting SNA trait values refers to a specific characteristic of the animal, in line with the definition above. Because the trait values depend on the method of calculations, it is crucial to clarify which type of social networks was used to calculate these traits, and that what we did here, in this study, in Line 111, where it is mentioned that “In this study, the edges were weighted based on the total duration of the reciprocal fighting behaviour between each pair of animals. SNA was used to give a score i.e., traits, for each animal to describe its role in the network, i.e., pen”.

Furthermore, SNA traits are not the only example where the values depend very much on the method of calculation. Other examples include Feed efficiency and Resilience, which are commonly defined as “traits” in the animal production literature.

 Lns 126-140: Lots of jargon in this paragraph, i.e., 'daily gain', 'on test', 'finishing period', 'feed efficiency, etc, which many people reading this journal may be unfamiliar with. Recommend defining or revising.

As the referee suggested, we modified the sentence by including the “during the entire life” in line 131. For the “finishing period”, we added the following sentence to Line 135 “the finishing period, (i.e., the phase in which pigs are fed until they reach market weight), was approximately from 70 to 172 days of age in the current experiment.

Ln 172: Does heritability refer to the genetic matrix described above, or was it calculated/defined in an undescribed manner? It's the first use of the term in the text, and I'm unsure how it fits into this section as it is currently used. Likewise, it appears to be used distinctly from genetic correlations in the remainder of the manuscript.

We modified the sentence as the reviewer suggested, to be “Posterior means and the standard deviations of the marginal distributions of genetic and environmental parameters of SNA and performance traits were estimated, in addition to the 95% highest posterior density intervals (HPD95%)”.

Lns 247-251: I'm glad authors noted the dependence of closeness centrality on social structure, particularly given the result - it was my first thought with the measure being included for analysis.

We thank the reviewer for the comment.

Lns 269-279: Adding to my point above, clustering coefficient and closeness centrality are dependent on social structure, which may not have developed sufficiently within the study window to find results. Happy the authors note this. I would also add here that weighted vs unweighted edges likely differ, but weighted based on duration, frequency, number of interactions, etc all change this measure, which again circles to my concern of referring to these SNA measures as 'traits'. Because we modify them using math, we need a lot more information before I would personally be comfortable aligning them with other, physical or biological traits.

Please see our response to the same point above.

Lns 307-312: Brilliant! Wonderful outcome and interpretation.

Thanks.

lns 320-323: Can we live in this world now!? I cant wait until we do.

We thank the reviewer for the comment.